# kNNDM CV: k-fold Nearest Neighbour Distance Matching Cross-Validation for map accuracy estimation

Jan Linnenbrink[1,*], Carles Milà[2,3*], Marvin Ludwig[1], and Hanna Meyer[1]

[1]Institute of Landscape Ecology, University of Münster, Münster, Germany
[2]Barcelona Institute for Global Health (ISGlobal), Barcelona, Spain
[3]Universitat Pompeu Fabra (UPF), Barcelona, Spain
[*]These authors contributed equally to this work.

**Correspondence:** Jan Linnenbrink (jan.linnenbrink@uni-muenster.de)

**Abstract.** Random and spatial Cross-Validation (CV) methods are commonly used to evaluate machine learning-based spatial prediction models, and the obtained performance values are often interpreted as map accuracy estimates. However, the appropriateness of such approaches is currently the subject of controversy. For the common case where no probability sample for validation purposes is available, in Milà et al. (2022) we proposed the Nearest Neighbour Distance Matching (NNDM) Leave-One-Out (LOO) CV method. This method produces a distribution of geographical Nearest Neighbour Distances (NND) between test and train locations during CV that matches the distribution of NND between prediction and training locations. Hence, it creates predictive conditions during CV that are comparable to what is required when predicting a defined area. Although NNDM LOO CV produced largely reliable map accuracy estimates in our analysis, as a LOO-based method, it cannot be applied to large datasets found in many studies.

Here, we propose a novel k-fold CV strategy for map accuracy estimation inspired by the concepts of NNDM LOO CV: the k-fold NNDM (kNNDM) CV. The kNNDM algorithm tries to find a k-fold configuration such that the Empirical Cumulative Distribution Function (ECDF) of NND between test and train locations during CV is matched to the ECDF of NND between prediction and training locations.

We tested kNNDM CV in a simulation study with different sampling distributions and compared it to other CV methods including NNDM LOO CV. We found that kNNDM CV performed similarly to NNDM LOO CV and produced reasonably reliable map accuracy estimates across sampling patterns. However, compared to NNDM LOO CV, kNNDM resulted in significantly reduced computation times. In an experiment using 4,000 strongly clustered training points, kNNDM CV reduced the time spent on fold assignment and model training from 4.8 days to 1.2 minutes. Furthermore, we found a positive association between the quality of the match of the two ECDFs in kNNDM and the reliability of the map accuracy estimates.

kNNDM provided the advantages of our original NNDM LOO CV strategy while bypassing its sample size limitations.

## 1 Introduction

Spatial predictive modeling using machine learning methods is commonly used in ecology and environmental sciences to predict variables sampled at a limited set of locations to new, unobserved locations (see e.g. van den Hoogen et al., 2019;

Sabatini et al., 2022; Moreno-Martínez et al., 2018; Hengl et al., 2017, for global studies). A key step in the spatial prediction workflow is the map accuracy assessment, i.e. the process whereby the quality of a prediction of a spatially-indexed variable in a finite and defined geographical area (e.g. a set of pixels forming a continuous surface) is estimated (Stehman et al., 2021; Wadoux et al., 2021). Although map accuracy assessment should ideally be done via design-based inference based on probability sampling (Wadoux et al., 2021), this is frequently not possible due to limited access to certain areas or expensive sampling methods (Martin et al., 2012). Instead, Cross-Validation (CV) methods are commonly used to estimate map accuracy. Previous studies, however, showed significant differences in map accuracy estimates depending on the type of CV being used, which initiated discussions on the appropriateness of these strategies (Wadoux et al., 2021; Meyer and Pebesma, 2022; Milà et al., 2022; Roberts et al., 2017; Ploton et al., 2020). Since CV is also typically used during model development (i.e during hyperparameter tuning (Schratz et al., 2019) and feature selection (Meyer et al., 2019)), reliable estimates of map accuracy are crucial to develop suitable prediction models.

Traditional CV methods that ignore the spatial structure of the data such as Leave-One-Out (LOO) or random k-fold CV (Hastie et al., 2009) have been found to provide reliable estimates of map accuracy when samples are randomly distributed within the entire prediction area, but not when they are clustered (Milà et al., 2022; Wadoux et al., 2021) or only cover parts of the prediction area (Meyer and Pebesma, 2021). As an alternative, spatial CV methods such as block CV (Wenger and Olden, 2012; Valavi et al., 2019; Roberts et al., 2017) or buffered-LOO CV (Telford and Birks, 2009; Le Rest et al., 2014; Brenning, 2022) are often used. Spatial CV methods are designed for geographical model transferability assessment, i.e. to test the ability of the model to make predictions for new geographic entities far away from the sampling areas, by designing a CV where independence between train and test data is sought (Roberts et al., 2017). Such strategies, however, have been found to underestimate map accuracy when reference data are regularly or randomly distributed within the entire prediction area. In some cases, this has even been reported for clustered samples (Wadoux et al., 2021; Milà et al., 2022). Recent proposals of methods for map accuracy estimation include sampling-intensity weighted CV, as well as model-based geostatistical approaches (de Bruin et al., 2022). However, the results of de Bruin et al. (2022) showed that these methods are not universal solutions and, depending on the sampling design, showed considerable over- or underestimation of the true map accuracy.

In our past work, we argued that the design of a CV method to provide a reliable estimate for map accuracy should be prediction-oriented, i.e. predictive conditions created during CV should resemble the conditions found when the trained model is applied to the prediction area (Milà et al., 2022; Meyer and Pebesma, 2022; Ludwig et al., 2023). Following this idea, in Milà et al. (2022) we considered predictive conditions in terms of geographical distances, and proposed the Nearest Neighbour Distance Matching (NNDM) LOO CV method, a prediction-oriented CV method based on spatial point patterns concepts. Briefly, NNDM aims to match the Empirical Cumulative Distribution Function (ECDF) of Nearest Neighbour Distances (NND) between test and train locations in the CV, to the ECDF of NND between prediction and training locations found during prediction.

In Milà et al. (2022) we showed that when standard LOO CV is used for reference data randomly distributed within the prediction area, the distribution of NND between test and training locations during CV is similar to the distribution of NND

between prediction and training locations (see Meyer and Pebesma, 2022, for similar results for random k-fold CV). In the case of clustered sampling designs, NND during LOO CV were generally much shorter than prediction NND which led to significant error underestimation (see also Ludwig et al., 2023). For regular samples, NND during LOO CV were found to be slightly longer than during prediction leading to slight error overestimation. With the newly developed NNDM LOO CV, we could produce comparable NND distributions in most sampling designs and provide more reliable estimates of map accuracy that can be used during model development or to indicate the accuracy of the predictions.

Even though NNDM LOO CV showed promising results across different sampling designs, prediction areas, and landscape autocorrelation ranges, the fact that it is a LOO-based CV method hinders its broader application given its high computational cost in medium and large datasets. To overcome this limitation, our aim is to extend the idea of NNDM LOO CV to a k-fold NNDM CV, hereafter kNNDM, that can be applied to larger datasets commonly found in ecology and the environmental sciences.

This article is organised as follows: after presenting our algorithm in section 2, in section 3 we reproduce the simulation study included in Milà et al. (2022) to assess the performance and runtime of kNNDM compared to other CV methods including NNDM LOO. In this simulation, we also explore the influence of the number of folds $k$ and the relationship between the quality of the match and the quality of the estimation of the map accuracy. As supplementary material, we provide a second simulation study which we also briefly describe in section 3. Finally, section 4 discusses the strengths and limitations of our method and suggests future lines of work.

## 2   Algorithm description

The objective of kNNDM is to find a k-fold configuration such that the distribution of NND between test and train locations during CV matches as close as possible the distribution of NND between prediction and train locations. In other words, kNNDM aims to create predictive conditions in terms of geographical distances that resemble those found when using the model to predict a certain area. To do so, we use a clustering approach to create a set of candidate fold configurations with different degree of spatial clustering, of which we choose the one that offers the best match between the two distributions. Before explaining the algorithm in detail, we define the different NND distribution functions used in kNNDM, as well as the statistic used to evaluate the different fold candidates.

As in the original NNDM LOO algorithm, in kNNDM we define nearest neighbour distance distribution functions by means of their NND ECDF, where $j$ is the index for training points, $i$ is the index for prediction points, and $r$ is a distance (Euclidean for projected coordinates and spherical for geographical coordinates):

- $\hat{G}_j(r)$ is the NND ECDF between test and train locations during LOO CV and expresses the proportion of training points that have another training point at a distance equal or lower than $r$:

$$\hat{G}_j(r) = \frac{1}{n_j}\sum_j 1\{d_j \leq r\} \quad \text{where} \quad d_j = \min_{k \neq j}\|x_j^{(j)} - x_k^{(j)}\|$$

– $\hat{G}_{ij}(r)$ is the NND ECDF between prediction and training locations and expresses the proportion of prediction points that have a sampling point at a distance equal or lower than $r$:

$$\hat{G}_{ij}(r) = \frac{1}{n_i} \sum_i 1\{d_{ij} \leq r\} \quad \text{where} \quad d_{ij} = \min \|x_i^{(i)} - x^{(j)}\|$$

     – $\hat{G}_j^*(r, \mathbf{L})$ is the NND ECDF between test and train locations during a CV defined in $\mathbf{L}$ and expresses the proportion of test points that have a training point at a distance equal or lower than $r$ during a given CV strategy. Note that $\mathbf{L} =$

$\{\mathbf{l}_1, \mathbf{l}_2, \ldots, \mathbf{l}_{n_j}\}$ is a list of sets $\mathbf{l}_j$ containing the indices of the samples to fit the model to when holding out observation $j$ during CV. Note that since in kNNDM we leave out data points in folds rather than one by one, $\mathbf{l}_j$ will be exactly the same for all samples belonging to the same fold:

$$\hat{G}_j^*(r, \mathbf{L}) = \frac{1}{n_j} \sum_j 1\{d_j^* \leq r\} \quad \text{where} \quad d_j^* = \min_{k \in \mathbf{l}_j} \|x_j^{(j)} - x_k^{(j)}\|$$

Another important component of our approach is how to measure the quality of the match between $\hat{G}_j^*(r, \mathbf{L})$ and $\hat{G}_{ij}(r)$

given a fold configuration. We do that by using the Wasserstein statistic (Dowd, 2020; Vaserstein, 1969), which compares the distribution of two samples by calculating the integral of the absolute value differences between the two ECDFs. In our case, we define W as the integral over the geographical distances $r$ of the absolute value differences between $\hat{G}_j^*(r, \mathbf{L})$ and $\hat{G}_{ij}(r)$:

$$W = \int |\hat{G}_j^*(r, \mathbf{L}) - \hat{G}_{ij}(r)| \mathrm{d}r$$

Small values of W indicate that the two ECDFs are similar, while W will be larger if they differ. To illustrate this point, we

show the calculation of the W statistic between $\hat{G}_j^*(r, \mathbf{L})$ and $\hat{G}_{ij}(r)$ for a random 10-fold CV and different sampling patterns (Figure 1). As shown in Meyer and Pebesma (2022), when samples are randomly distributed within the prediction area, the distribution of NND during the random 10-fold CV $\hat{G}_j^*(r, \mathbf{L})$ resembles the distribution of NND during prediction $\hat{G}_{ij}(r)$, and therefore the value of W is small. However, in the presence of clustered samples, random k-fold CV NND are shorter than prediction NND, and thus $\hat{G}_j^*(r, \mathbf{L}) > \hat{G}_{ij}(r)$, resulting in a large W value. The opposite occurs when training samples follow

a regular sampling pattern, also leading to a larger W statistic compared to random sampling.

Now that the definitions of the NND functions and the W statistic have been stated, we proceed with the explanation of the kNNDM algorithm (Figure 2). The first step is to test whether the training points are clustered with respect to the prediction points; to do that we compute $\hat{G}_j(r)$ and $\hat{G}_{ij}(r)$, and then test the null hypothesis $G_j(r) \leq G_{ij}(r)$ vs. the alternative $G_j(r) > G_{ij}(r)$ with a one-sided Kolmogorov–Smirnov (KS) two-sample test (Conover, 1999). If the null hypothesis is not

rejected, the algorithm returns a random k-fold CV since it is the appropriate option for random and regular samples (Meyer and Pebesma, 2022; Wadoux et al., 2021; de Bruin et al., 2022). If, however, the null hypothesis is rejected (p-value < 0.05), we proceed to cluster the training points based on their coordinates into a range of $q_i \in Q$ clusters, where $Q$ is defined as an integer sequence of length 100 ranging between $k$ and $N$ (the total number of training points) equally-spaced in the logarithmic scale and back-transformed, to try configurations with a small number of clusters more intensively. Currently, hierarchical and

k-means clustering methods are implemented.

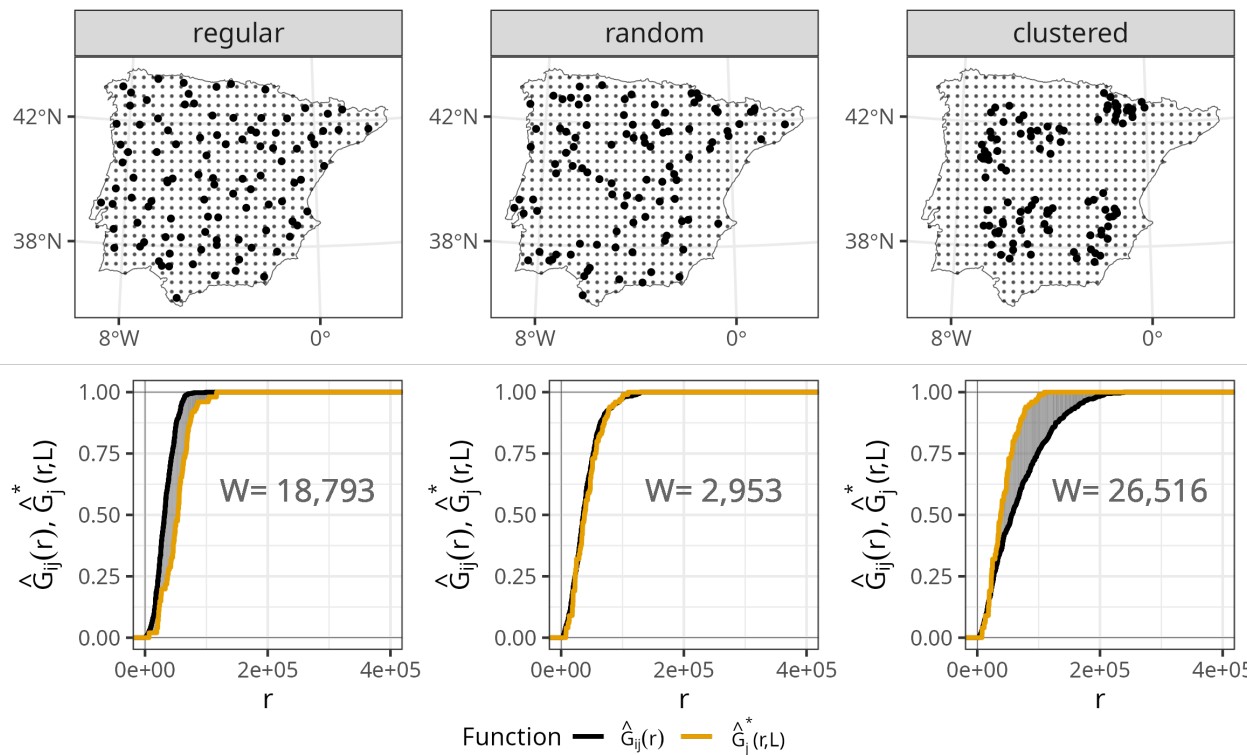

**Figure 1.** Top row: prediction points (regular grid) and training points with different spatial distributions (bold), simulated for visualization purposes only. Bottom row: NND ECDF between training and test locations during 10-fold random CV ($\hat{G}_j^*(r, \mathbf{L})$, orange) and NND ECDF between prediction and training locations ($\hat{G}_{ij}(r)$, black) corresponding to each of the sampling distributions in the top row. The shaded grey area corresponds to the W statistic, whose value is printed in the plots.

Next, for every configuration where $q > k$, we merge the resulting $q$ clusters into the final $k$ folds along the first principal component of the coordinates of the training points to prevent contiguous clusters in space to be in the same fold. Briefly, we compute the first principal component of the training points' coordinates to capture the direction with the most spatial variability, project the $q$ cluster centroids into that first component and order them according to it, and finally merge $q$ clusters into $k$ folds by giving different fold levels to contiguous clusters in that dimension. Large clusters with a proportion greater than $1/k$ of the training data are not merged to keep fold balance. Once this procedure is completed, we compute $\hat{G}_j^*(r, \mathbf{L})$ and W for each fold configuration candidate and select the one with the smallest W, i.e. the one that provides the best match between $\hat{G}_j^*(r, \mathbf{L})$ and $\hat{G}_{ij}(r)$.

As an illustration of how kNNDM works in cases where samples are clustered within the prediction area, Figure 3 shows a 2-fold kNNDM CV configuration for different number of clusters $q$, their respective NND ECDF functions, and the W statistic between $\hat{G}_j^*(r, \mathbf{L})$ and $\hat{G}_{ij}(r)$ assessing the match. A low number of clusters leads to a strong partition of the geographical space and long NND between test and train points during CV, which are actually longer than NND encountered when predicting

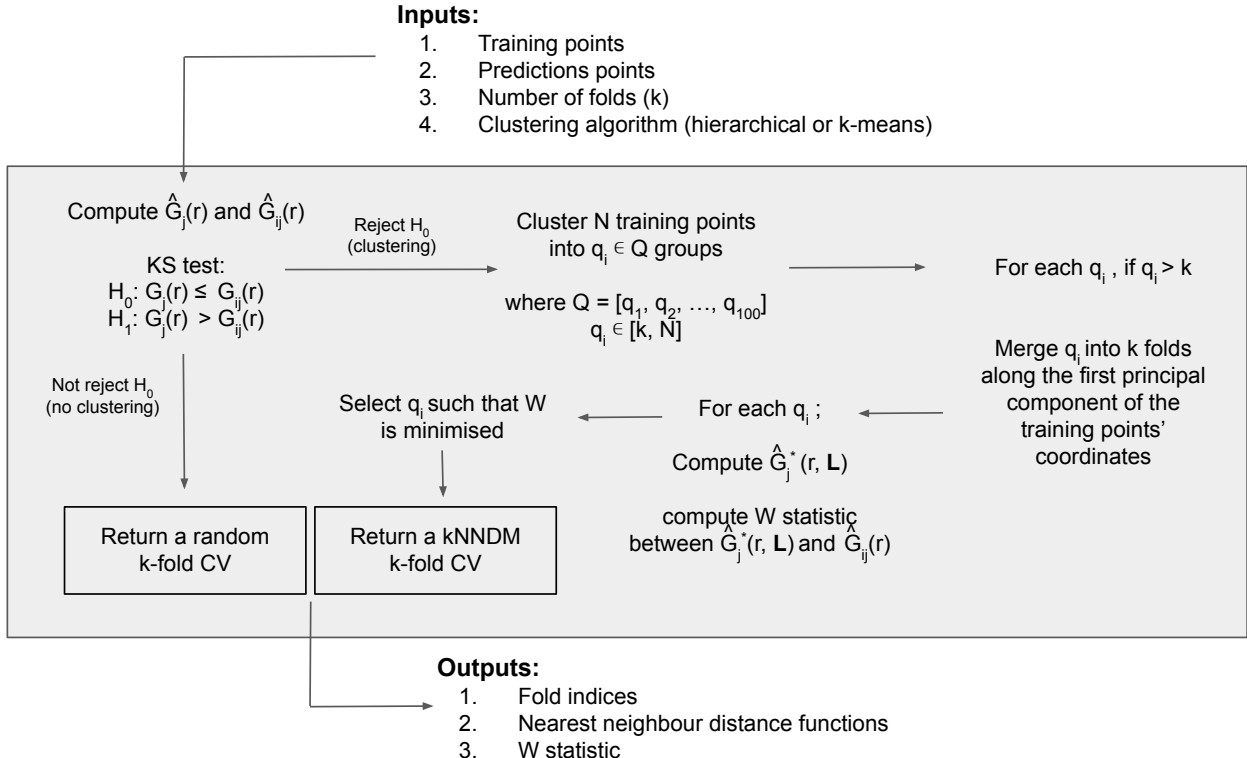

**Inputs:**
1. Training points
2. Predictions points
3. Number of folds (k)
4. Clustering algorithm (hierarchical or k-means)

Compute $\hat{G}_j(r)$ and $\hat{G}_{ij}(r)$

KS test:
$H_0: G_j(r) \leq G_{ij}(r)$
$H_1: G_j(r) > G_{ij}(r)$

Reject $H_0$
(clustering)

Cluster N training points into $q_i \in Q$ groups

where $Q = [q_1, q_2, ..., q_{100}]$
$q_i \in [k, N]$

For each $q_i$, if $q_i > k$

Not reject $H_0$
(no clustering)

Select $q_i$ such that W is minimised

For each $q_i$;

Merge $q_i$ into k folds along the first principal component of the training points' coordinates

Compute $\hat{G}_j^*(r, \mathbf{L})$

Return a random k-fold CV

Return a kNNDM k-fold CV

compute W statistic between $\hat{G}_j^*(r, \mathbf{L})$ and $\hat{G}_{ij}(r)$

**Outputs:**
1. Fold indices
2. Nearest neighbour distance functions
3. W statistic

**Figure 2.** Workflow of the kNNDM algorithm.

from all reference data. As the number of clusters increases, $\hat{G}_j^*(r, \mathbf{L})$ gets closer to $\hat{G}_j(r)$, i.e. the NND ECDF encountered during LOO CV. In this example, the kNNDM algorithm would select the configuration with $q = 21$ since it minimises the W

statistic, i.e. provides the best match.

As practical considerations, in our implementation we have provided the possibility, as an alternative to the prediction points input, of supplying a polygon defining the prediction area, from which prediction points are sampled internally. Moreover, we have included a balancing parameter for the maximum single fold size allowed that discards non-compliant fold candidates. Regarding computational performance, our algorithm benefits from using projected coordinates since fast nearest neighbourhood

searches in Euclidean space can be done using the FNN package (Beygelzimer et al., 2022). Finally, when using kNNDM we recommend computing accuracy statistics such as the coefficient of determination ($R^2$), the Root Mean Square Error (RMSE), or the Mean Absolute Error (MAE) in the stacked out-of-sample predictions rather than performing an average of the statistics computed in each of the folds, since the resulting folds can be unbalanced and $\hat{G}_j^*(r, \mathbf{L})$ is constructed using all CV folds simultaneously (Meyer et al., 2023).

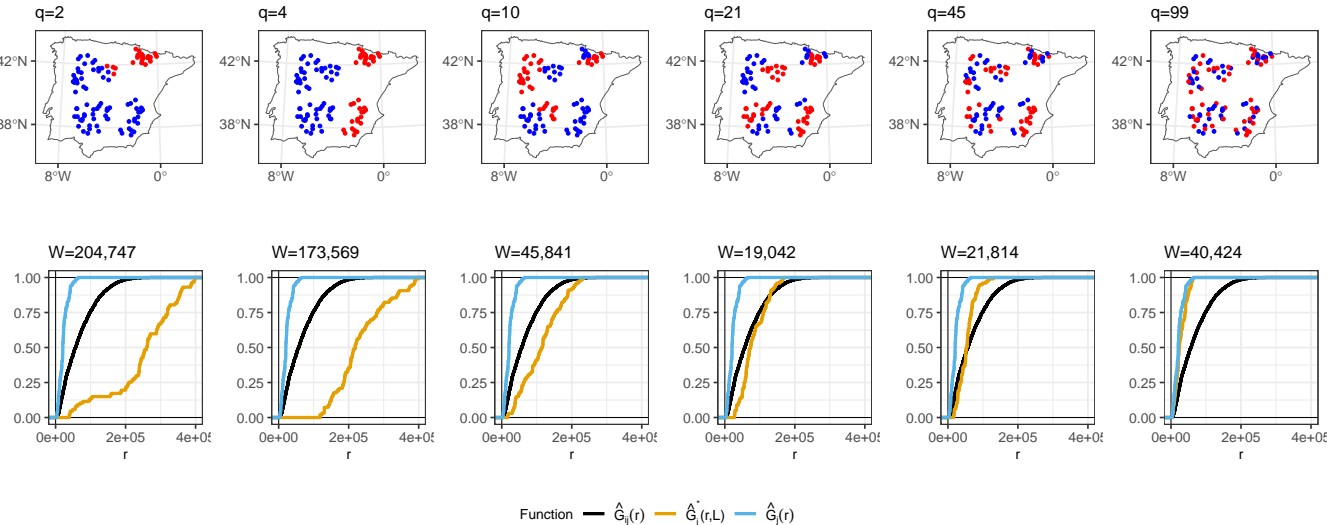

**Figure 3.** Top row: kNNDM with k=2 (red and blue points) for several number of clusters $q$. Prediction points consist of a regular grid (not shown) spanning the whole polygon. Points were simulated for visualization purposes only. Bottom row: NND ECDF between training locations during LOO CV ($\hat{G}_j(r)$, blue), between test and train locations during kNNDM CV ($\hat{G}_j^*(r, \mathbf{L})$, orange), and between prediction and training locations ($\hat{G}_{ij}(r)$, black) corresponding to each CV configuration in the top row.

## 3 Simulation studies

### 3.1 Algorithm performance for map accuracy estimation

To investigate the performance of kNNDM CV and how it compares to the original NNDM LOO CV, we used the same simulation as in our previous work (see Milà et al. (2022) for a complete description). Briefly, we generated a virtual response surface, i.e. a spatially-indexed continuous variable between 0 and 1, using a selection of WorldClim bioclimatic variables for the Iberian peninsula (Figure 4A) and the `virtualSpecies` R package (Leroy et al., 2015). Next, we simulated training locations with five different distributions and a sample size of 100 (Figure 4B). We performed 100 iterations of the sampling simulation and, in each of them, we extracted the predictor (bioclimatic variables) and response data at the sampling points' locations and fitted Random Forest (RF) regression models, resulting in a total of 500 fitted models. RF hyperparameters were not tuned and default values were used in all simulations to shorten computation time.

Each fitted RF model was used to predict the response in the entire prediction area, from which the true map accuracy was calculated (RMSE, MAE and $R^2$). Additionally, we used the following CV methods: 1) random 10-fold CV, 2) spatial 10-fold CV via k-means clustering (Brenning, 2012), 3) the original NNDM LOO CV and 4) 10-fold kNNDM CV. In contrast to the original simulation in Milà et al. (2022), here we matched all distances in the prediction area during NNDM rather than up to

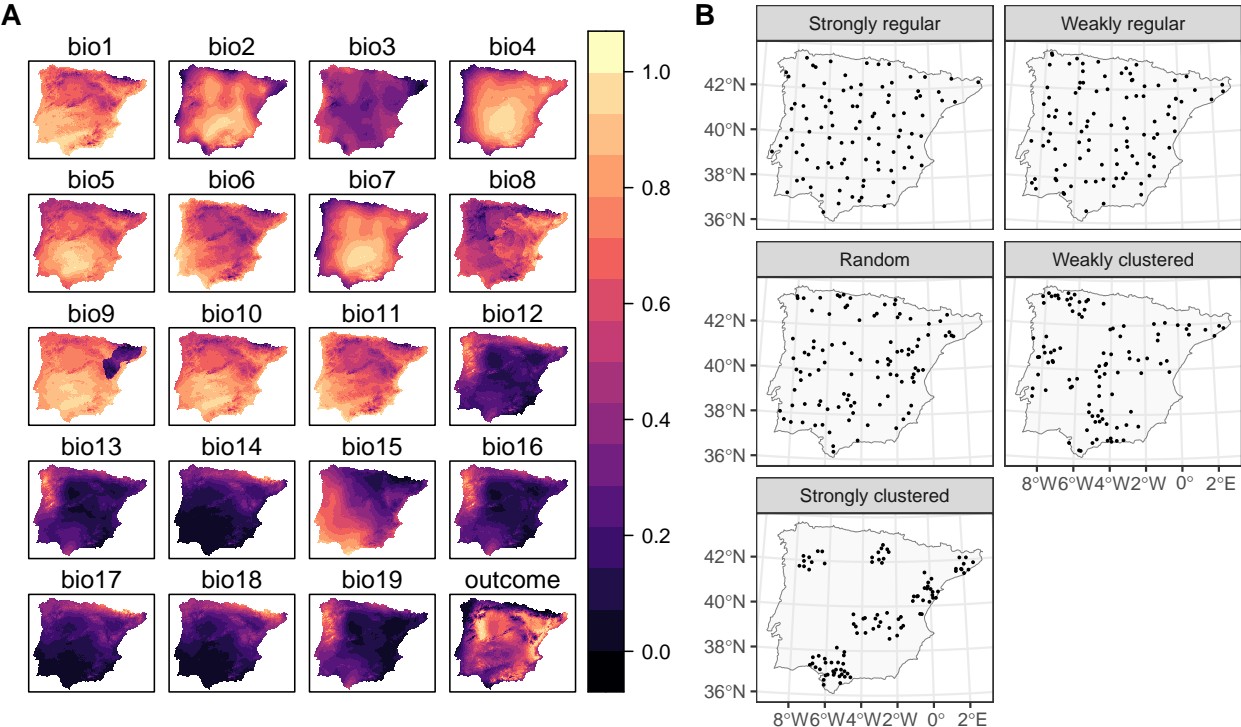

**Figure 4.** Data used in the simulation: A) bioclimatic covariates and response (all linearly stretched to [0,1] for visualization purposes); and B) example of one iteration of the sample simulation. Figure reproduced as in Milà et al. (2022).

a certain range estimated from the data. In order to interpret results, we subtracted the true map accuracy metrics from each of the CV estimates to assess their performance (Figure 5).

kNNDM CV yielded reliable error estimates across all sampling patterns we considered, which were similar and in some cases even more accurate than those estimated via NNDM LOO CV (Figure 5). Variability of the differences was larger in kNNDM than in NNDM LOO CV for strongly clustered samples. Random 10-fold CV produced reliable estimates under random sampling patterns but failed for clustered data. The spatial 10-fold CV overestimated the mapping error except for the RMSE in the strong clustering scenario and had the largest variability.

### 3.2 Relationship between the quality of the match and the quality of the map accuracy estimate

In order to investigate the relationship between the quality of the match in kNNDM and the quality of the map accuracy metrics, we performed a second simulation using the same response and predictors and 100 iterations of our first simulation (section 3.1). However, in this second simulation, we 1) added two more extreme clustered sample configurations to extend the range of possible W values, 2) only used kNNDM CV, 3) did not check for clustering as a first step in kNNDM, i.e. we applied the clustering approach to all samples regardless of their distribution, and 4) we kept all candidate fold configurations $q_i \in Q$

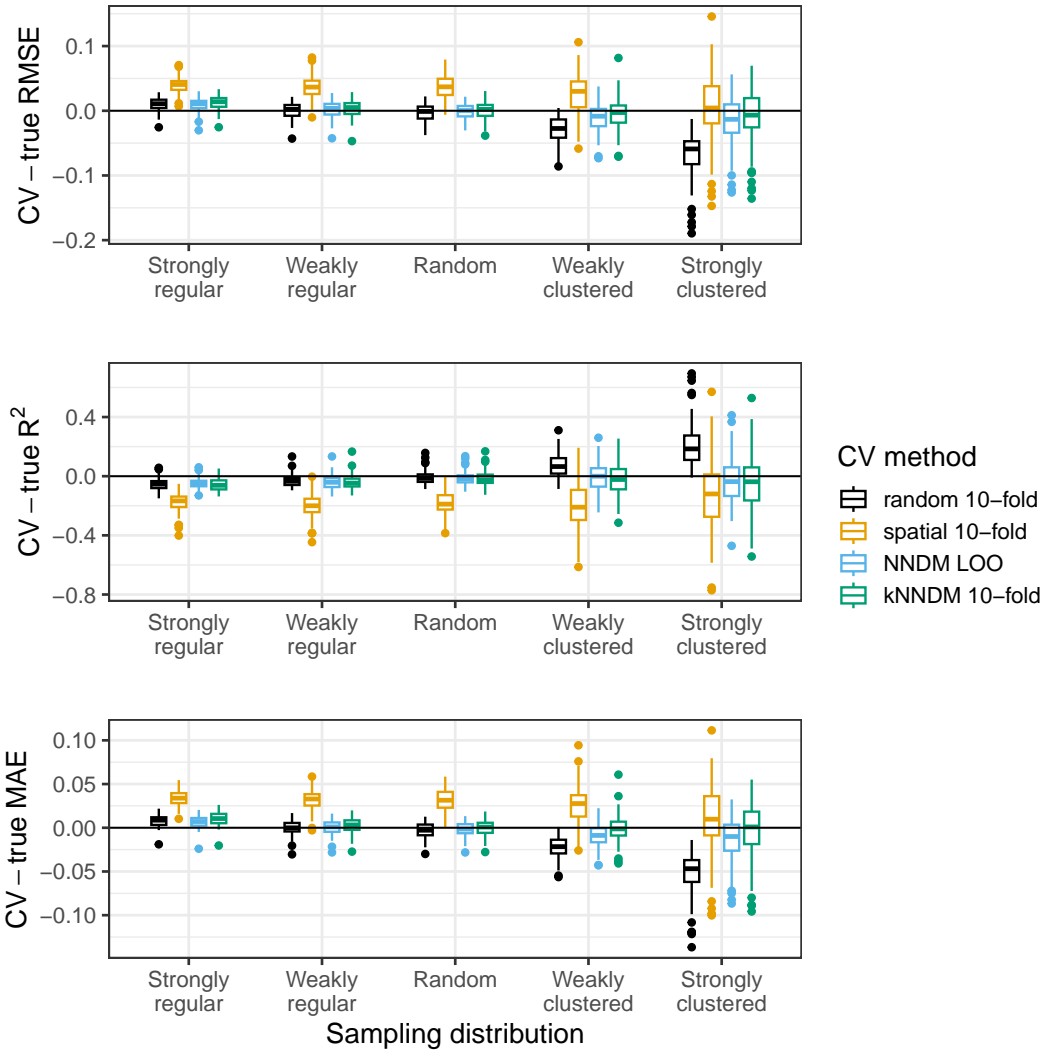

**Figure 5.** Differences between cross-validated and true RMSE, R$^2$, and MAE by sampling distribution and CV method for the simulated virtual response variable.

considered within kNNDM and their respective values of the W statistic, rather than just the one yielding the lowest W. We used all of these candidate CV splits to calculate CV map accuracy statistics, and computed the absolute value difference with respect to the true value of the map accuracy statistic. We then plotted these absolute value differences against the corresponding W statistic and fitted a linear regression to summarise the trend (Figure 6).

The relationship between the absolute value differences between CV and true map accuracy statistics (Figure 6) and W showed that, for all three statistics considered, a poor match between $\hat{G}_j^*(r, \mathbf{L})$ and $\hat{G}_{ij}(r)$ indicated by a greater W statistic led

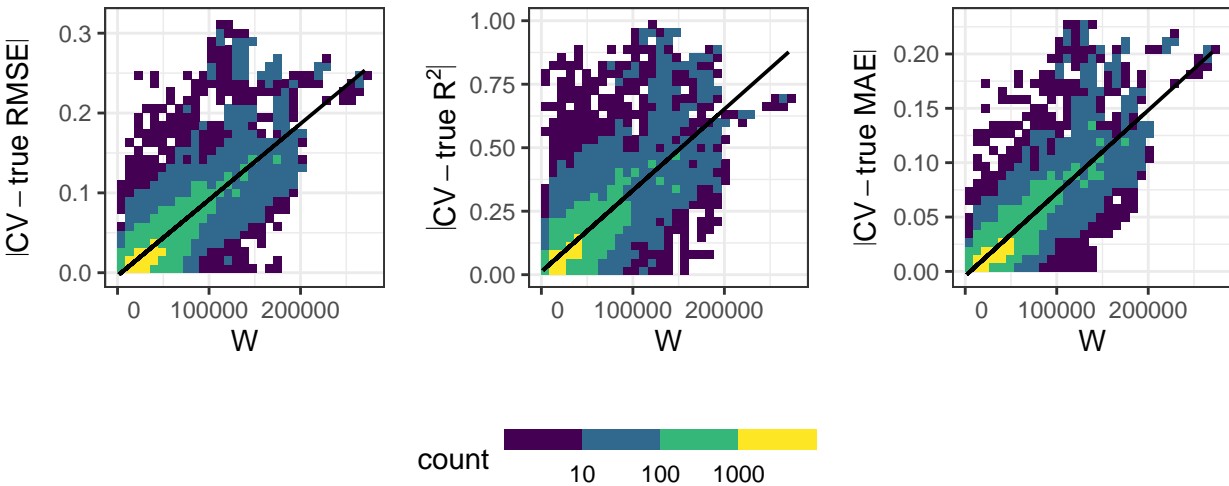

**Figure 6.** Relationship in 10-fold kNNDM between the absolute value difference between the CV and true map accuracy statistics (y-axis) and W statistic (x-axis) for the RMSE (left), $R^2$ (centre), and MAE (right) statistics. Colour represents the data point density. The black line shows the linear regression fit. The $R^2$ values of the regression models are 0.66, 0.6 and 0.73 for the RMSE, $R^2$, and MAE, respectively.

to poor estimates of the true map accuracy, while the true map accuracy could be better estimated for well-matching functions. This positive association was linear for all three statistics with at least 60% explained variance.

## 3.3 Influence of the number of folds

The choice of $k$ can influence the performance of kNNDM to a certain extent since it dictates the maximum clustering that can be achieved. To investigate the influence of $k$, we repeated the workflow described in section 3.1 but only employed kNNDM CV using an even integer sequence $k \in [2, 4, 6, ...20]$. In each of the 100 iterations, we calculated the true and estimated error metrics, as well as the quality of the match between the ECDF of NND between CV folds ($\hat{G}_j^*(r, \mathbf{L})$) and the ECDF of NND between prediction points and sample points ($\hat{G}_{ij}(r)$) as measured by the Wasserstein statistic (W). With the resulting statistics, we plotted the distribution of the differences between the estimated and true RMSE, MAE, and $R^2$ as well as the W statistic for each number of $k$ (Figure 7).

Our results indicated that a larger number of folds resulted in better matches for regular and random samples but worse for strongly clustered designs. While for regular and random samples this translated into increasingly accurate map accuracy estimates for larger number of folds, for clustered data, the number of folds with the smallest W, i.e $k = 2$ was overly pessimistic and $k$=4 or 6 had actually a better performance.

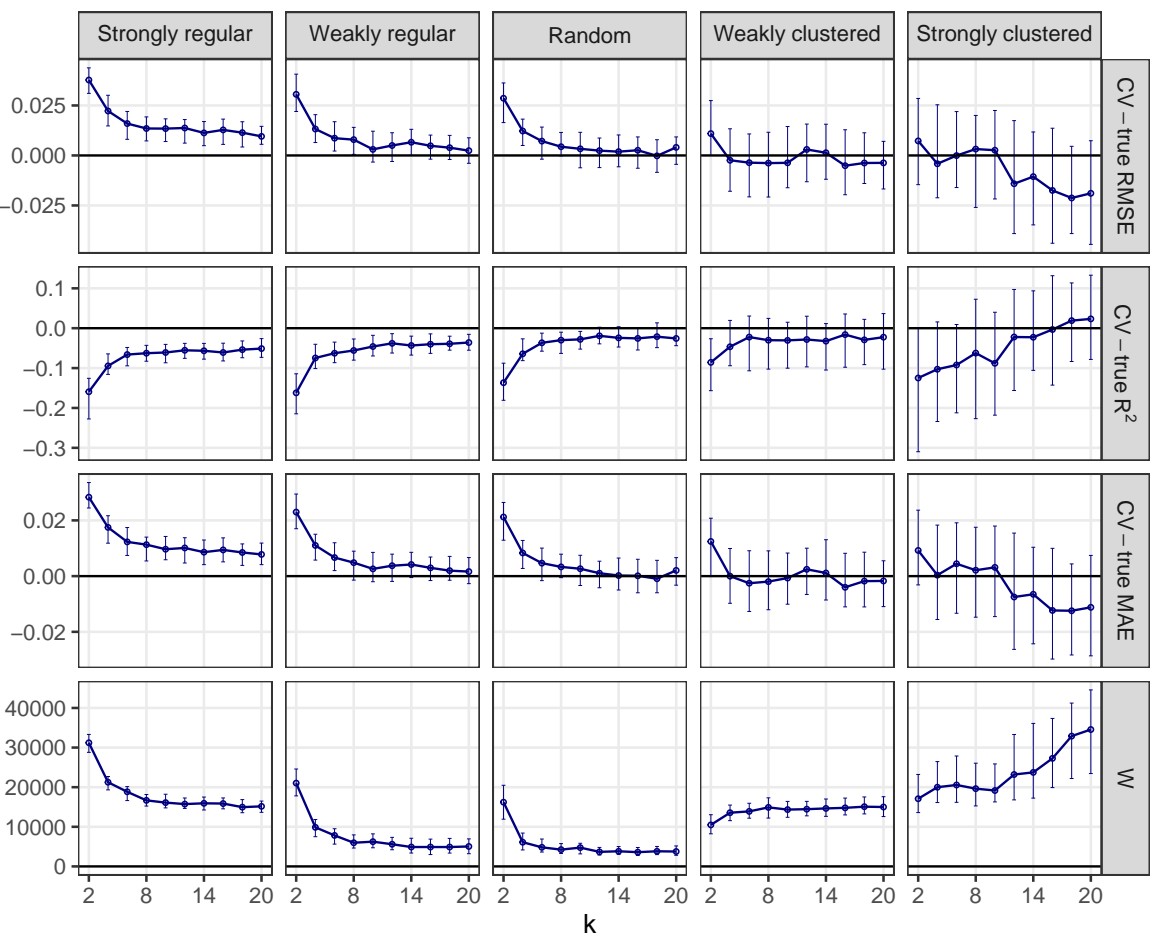

**Figure 7.** CV error estimates for kNNDM CV with different numbers of $k$ (first three rows). The respective W statistic is shown in the fourth row. Points indicate median values, while error bars show the first and third quartile.

## 3.4 Run-time analysis

Since our goal was to propose a computationally feasible alternative to NNDM LOO CV for large datasets, we performed a run-time analysis to quantify the speed gains of kNNDM CV compared to NNDM LOO CV. We separately quantified the

time spent on 1) finding the optimal CV split (i.e., running the NNDM LOO and kNNDM algorithms), 2) repetitively fitting the model according to the CV configuration, and 3) the total run-time, i.e. the sum of 1 and 2. We did that using the same simulation framework as in section 3.1 but with 50 different sample sizes, ranging from 100 to 4000 training points. We only used the strongly clustered and the random sampling designs for computational reasons. We used a maximum of 4000 training points, since the computational time exceeded one week per run for NNDM LOO CV. The analysis was carried out using a

high-performance computation cluster utilizing up to 1.5 GB of RAM for each run using the Intel® Xeon® Gold 6140 CPU.

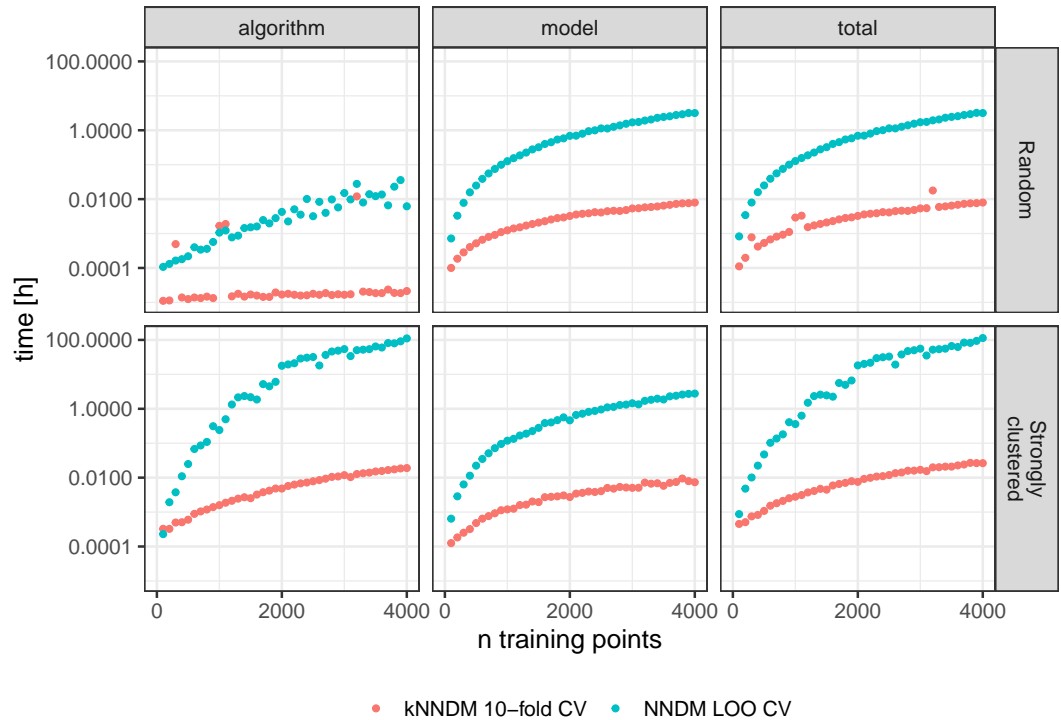

**Figure 8.** Differences in computational time (log-scaled) between 10-fold kNNDM CV (red) and NNDM LOO CV (blue) for two different sample designs (rows).

The run-time analysis showed large speed gains of kNNDM CV compared to NNDM LOO CV under all tested sample designs (Figure 8). For the random sampling design, the kNNDM algorithm was especially fast due to the prior test for clustering in the training data. This test returns a simple random k-fold CV when no clustering is detected, which is much faster than running the entire kNNDM algorithm (see section 2). Only in four cases the test did not return a random CV, and in those four cases the computational times were longer (red outliers in Figure 8).

NNDM LOO was slower compared to kNNDM when the training data were randomly distributed, and much slower when they were clustered (Figure 8). For a sample size of 4,000, kNNDM CV reduced the time spent on fold assignment and model training from 3.2 hours to 30 seconds for random samples and from 4.8 days to 1.2 minutes for clustered samples, as compared to NNDM LOO CV. Furthermore, the computational time of NNDM LOO CV increased exponentially with increasing sample size. This pattern arises from both the architecture of the NNDM LOO CV algorithm (Figure 8 left column), and from the difference between of LOO CV and k-fold CV in terms of model fitting, since NNDM LOO CV requires training $k = N$ models while the number of models trained during k-fold CV is usually much smaller, in this case $k = 10$ (Figure 8 middle column).

## 3.5 Additional simulation study

To test the robustness of our results, we tested the performance of kNNDM CV in a second simulation using a real-world dataset described in detail in appendix 4 and used in the study by de Bruin et al. (2022). Briefly, we modelled above-ground biomass in Europe using different sampling distributions ranging from regular to strongly clustered (Figure A1).

Results for the second simulation broadly agreed with the first simulation, although we observed some differences worth pointing out. 10-fold kNNDM reliably estimated the true RMSE in all designs except in the strongly clustered, where similar to spatial CV it resulted in overly optimistic estimates (Figure A2). The relationship between the absolute value difference between CV and true RMSE was positive but only explained a 28% explained of the total variance (Figure A3). While large numbers of folds resulted in a better match as expressed by W and more accurate RMSE for regular and random designs (Figure A4), in strongly clustered designs the opposite was observed. For weak clustered results, $k = 4$ had the lowest W statistic although RMSE were also well estimated for larger values of $k$.

## 4 Discussion

In this work, we propose a new prediction-oriented CV strategy for map accuracy estimation named kNNDM that takes into account the geographical prediction space of the model. kNNDM extends the ideas of NNDM LOO CV to a k-fold CV strategy that can be used for medium and even large reference datasets to estimate map accuracy in absence of probability sampling test data. In the simulation study, kNNDM performed similarly to NNDM LOO CV and produced reasonably reliable map accuracy estimates for most sampling patterns. Thus, kNNDM provided the advantages of our original NNDM LOO strategy while bypassing its sample size limitations. Small differences between NNDM LOO and kNNDM CV can be attributed to the different ways to match the distributions as well as the different hold-out sample size.

Similar to other studies (e.g. Wadoux et al., 2021; de Bruin et al., 2022), we observed that random k-fold CV returned reliable estimates of map accuracy with randomly distributed samples within the prediction area, while being overly optimistic when samples were clustered. Also in agreement with other studies, we found that spatial CV methods that do not take into account the geographical prediction space tended to be overly pessimistic even with clustered samples within the prediction area (Wadoux et al., 2021; de Bruin et al., 2022; Milà et al., 2022), for example as a result of block sizes that do not match the prediction task. A unique finding of our study that deserves special attention is the positive association we found between the W statistic measuring the quality of the match of the NND ECDFs during CV and prediction, and the quality of the estimation of the map accuracy statistics. This association was strong in our first simulation with a national scale, supporting our suggestion to design CV strategies that try to match the predictive conditions of the models in terms of geographical NND. That said, this relationship was weaker in the second simulation, where the study area had a continental scale. This suggests that other factors such as distances in the feature space may also play a role in the performance of CV map accuracy estimates.

Our experiments showed that the number of folds can have an impact on the performance of kNNDM. For randomly and regularly-distributed samples, $k$ needs to be sufficiently large ($k \geq 10$) to provide accurate estimates of map accuracy. The same finding applies to random k-fold CV, to which kNNDM generalises for random and regular samples. We attribute this to

the fact that, when the number of clusters is small, neighbouring samples can be put in the same fold with a probability that increases with smaller $k$. On the other hand, for severely clustered samples, a smaller number of folds may be beneficial as $k$ determines the maximum clustering that can be achieved when the geographical space is partitioned in $k$ contiguous blocks. The suitability of a smaller number of folds was indicated by a higher quality of the match shown by the Wasserstein statistic. Comparing the suitability of different fold configurations via the Wasserstein statistic can be used for guidance when choosing the number of folds. Nonetheless, in clustered settings where W indicates that the best match is achieved by setting a very low k such as $k=2$ (see Figure 7), we recommend taking a larger fold size such as 4 or 5 since the amount of bias expected with two folds due to large parts of the training data left out is expected to be large (Kohavi et al., 1995) and is likely the reason we observe better results for $k=4$ or 6 in our experiments.

Even though kNNDM overcomes the sample size limitations of NNDM LOO CV, there are still limitations of the approach. First, the flexibility of the matching in kNNDM is lower than in NNDM LOO CV, since every observation must be assigned to a fold. Moreover, it is also possible that the range of NND observed during prediction is different than the range of NND between training points, which might make the match impossible for some distances. This is especially the case when the prediction area is different from the training area (i.e. complete model transfer). Because of these reasons, the match between CV and prediction NND ECDF in kNNDM may not always be possible and an inspection of the NND ECDF like in Figure 1 should always be conducted. Similarly, if training data are very clustered within the prediction area as in the strongly clustered design of the second simulation, kNNDM may still fail to offer a CV configuration that matches the predictive conditions. In that case, we recommend users to allow for a greater maximum fold size, or ask for a lower fold number $k$, to account for potentially larger clusters. Furthermore, in cases where this is still not sufficient, we recommend restricting the prediction area to the area of applicability of the model (AOA, Meyer and Pebesma (2021)) to limit the effects of feature extrapolation. Secondly, both kNNDM and NNDM LOO CV algorithms are currently solely based on the geographical space; therefore, if the feature distribution between the training and prediction locations is very different, a feature-based CV strategy might be more appropriate (Roberts et al., 2017). For example, Wang et al. (2023) recently developed a CV method that considers both the geographic and feature space, although it does not consider the prediction domain and predictive conditions of the model. Thirdly, NNDM-based CV methods do not address the small error overestimation for regular samples we found in our simulations, so map accuracy estimates will tend to be slightly conservative in such cases. Fourthly, NNDM-based methods are purely based on geographical distances and ignore the location of the training points or the direction of the distances, which can be problematic if non-stationarity or anisotropy of the errors are present (Brenning, 2022). Fifthly, the CV error estimate obtained by kNNDM is only reasonable if the prediction area does not change when the model is deployed. If the prediction area changes significantly, re-evaluation might be required. Also, when the prediction area is unknown prior to model training, kNNDM cannot be used.

Possible future points for investigation regarding kNNDM include a simulation study comparing newly proposed CV-based map accuracy estimation methods (de Bruin et al., 2022) as well as feature-based CV methods (Roberts et al., 2017) in a larger variety of scenarios, also including classification problems, 2) implementing a genetic algorithm that minimizes the W statistic directly as a function of CV folds, 3) exploring the extension of kNNDM to feature space, and 4) investigating how kNNDM CV

can affect feature selection (Meyer et al., 2019), hyperparameter tuning (Schratz et al., 2019), and model applicability (Meyer and Pebesma, 2021). A further possible extension to the kNNDM algorithm is the exclusion of training points during CV, which might help to achieve a better match in strongly clustered designs without the need to increase fold sizes. Furthermore, it might be beneficial to develop and integrate a one-sided Wasserstein test instead of using the Kolmogorov-Smirnov two-sample test to test whether the training points are clustered as the first step of the algorithm, since the Wasserstein test has a greater power than Kolmogorov-Smirnov's (Dowd, 2020) and would be also be more consistent with the rest of the algorithm, which also uses the Wasserstein statistic.

Finally, we would like to emphasize again that NNDM and kNNDM CV do not replace established strategies to estimate map accuracy via design based inference (as outlined in Wadoux et al., 2021), which should always be preferred. Nonetheless, prediction-oriented CV methods such as NNDM LOO or kNNDM CV that consider the prediction objectives of the model can be used to implement a measure of map accuracy during model development or, in the absence of a probability sample, to estimate the map accuracy of a given model.

*Code availability.* All simulations were carried out in R v 4.2.1 (R Core Team, 2022). The most important packages used include `twosamples` (Dowd, 2022) for efficient calculation of the W statistic, `doParallel` (Corporation and Weston, 2022) for parallelization, `tidyverse` (Wickham et al., 2019) for data manipulation and `ggplot2` (Wickham, 2016) for data visualization. We used `sf` (Pebesma, 2018) for vector data operations, `terra` (Hijmans, 2023) for raster data operations and `caret` (Kuhn, 2022) for model fitting. NNDM LOO as well as the newly suggested kNNDM algorithms are implemented in the `CAST` package v 0.7.2 (Meyer et al., 2023). The code to perform the analysis and generate the figures included in the article is available at https://doi.org/10.6084/m9.figshare.23514135.v1, where the packages and the versions used for the simulations are listed.

**Appendix A: Additional simulation study**

This supplement describes a second simulation study used to assess the robustness of our findings. We used the example presented in de Bruin et al. (2022) where a set of 22 predictor variables (see Appendix A of de Bruin et al. (2022) for a list of the predictor variables and data sources) was used to predict Above-Ground Biomass (AGB, Santoro and Cartus (2021)) in Europe using different sampling designs (Figure A1). This dataset was used to test the same hypotheses as in our main simulation: 1) to test the ability of kNNDM to estimate true map accuracy, 2) to assess the relationship between the quality of the match and the quality of the map accuracy estimate, and 3) to explore the impact of the number of folds on the performance of kNNDM. All methods and visualizations follow our main simulation as much as possible to allow for comparison.

To test the ability of kNNDM to estimate the map accuracy (first objective), Random Forest (RF) models with fixed default hyperparameters were trained on 5,000 sample points generated by five sampling designs ranging from regular to clustered (Figure A1) as in de Bruin et al. (2022). Then, their predictive performance (measured as the Root Mean Square Error, RMSE) was estimated using different types of Cross-Validation (CV) and was compared to the true RMSE for the entire surface. Namely, we assessed the performance of random 10-fold CV, spatial 10-fold CV based on kmeans clustering of the geographical coordinates (Brenning, 2012), and 10-fold kNNDM CV. We did not include NNDM LOO CV since it was computationally not feasible (see Figure 7 in the main manuscript for a run-time analysis). While in the original study by de Bruin et al. (2022) the CV split was repeated 100 times per sample distribution and CV error estimates were averaged, we only used one repetition to shorten computational times. For a more detailed description of the simulation, we refer the reader to de Bruin et al. (2022). We subtracted the estimated and the true RMSE and plotted the distribution of these differences by CV method (Figure A2).

To assess the relationship between the quality of the match and the quality of the map accuracy estimate (second objective), we repeated the workflow described in the first objective but 1) only used kNNDM CV, 2) did not check for clustering as a first step in kNNDM, i.e. we applied the clustering approach to all samples regardless of their distribution, and 3) we kept all candidate fold configurations $q_i \in Q$ considered within kNNDM and their respective values of the W statistic, rather than just the one yielding the lowest W. We used all of these CV configurations to estimate map accuracy, and computed the absolute value difference of the estimates and the true value. We then plotted these differences against the corresponding W statistic and fitted a linear regression to estimate the relationship between the two (Figure A3).

To explore the impact of the number of folds on the performance of kNNDM (third objective), we repeated the workflow described in the first objective but employed kNNDM CV using an even integer sequence $k \in [2, 4, 6, ..., 20]$. In each of the 100 iterations, we calculated the true and estimated RMSE, as well as the quality of the match as measured by the Wasserstein statistic (W). With the resulting statistics, we plotted the distribution of the differences between the estimated and true RMSE as well as the W statistics (Figure A4).

**Figure A1.** Data used in the AGB simulation: A) shows a subset of the 22 predictors, as well as the response raster. Rasters were linearly stretched to [0,1] for visualization purposes. B) shows one iteration of the different sampling designs. Data from de Bruin et al. (2022).

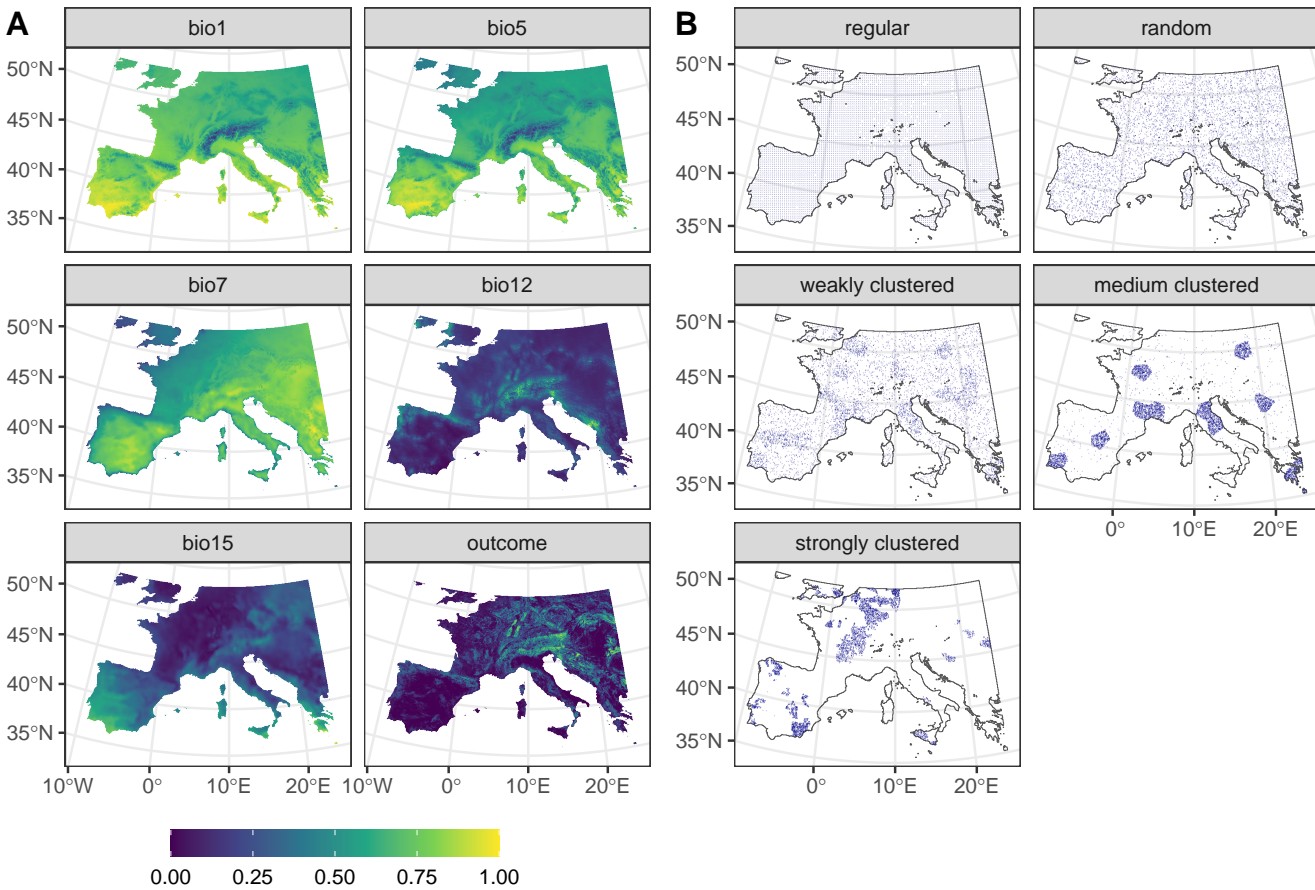

**Figure A2.** Differences between cross-validated and true RMSE. kNNDM [tuned] refers to the kNNDM split that yielded the lowest W statistic among different number of folds $k \in [2, 4, 6, ..., 20]$.

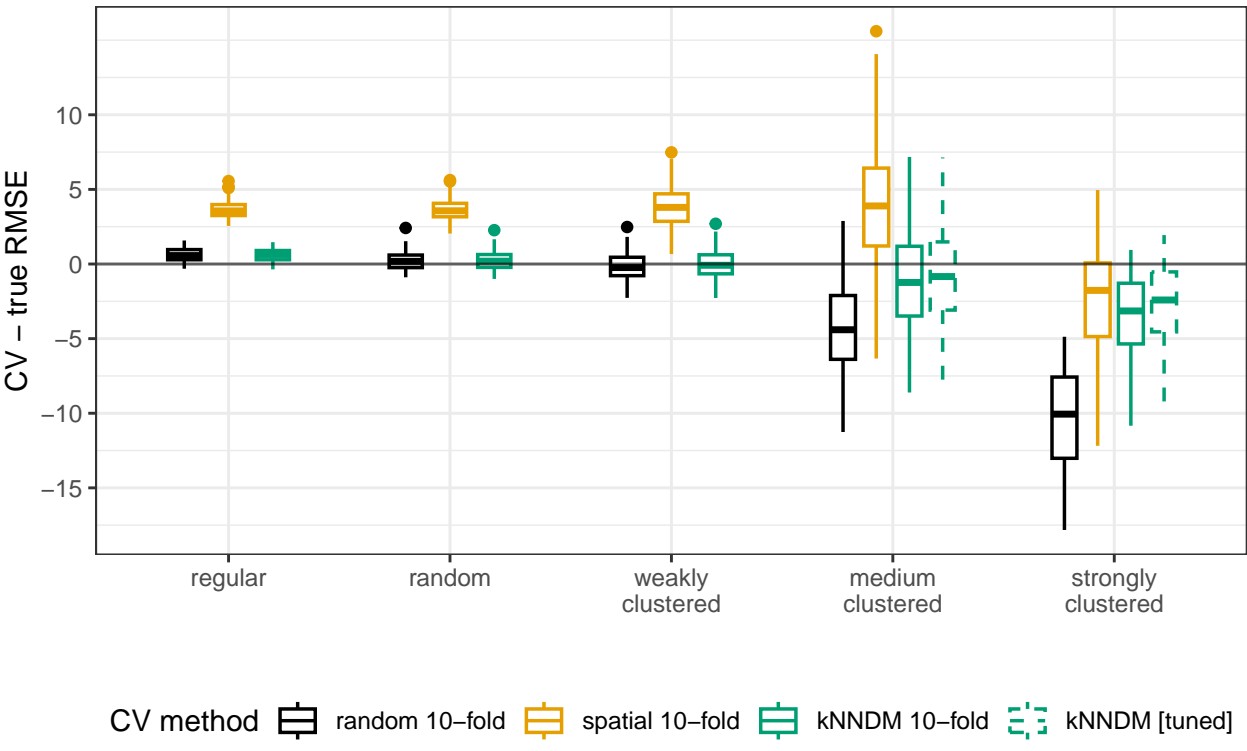

**Figure A3.** Relationship between the absolute value difference between the CV and true RMSE (y-axis) with the W statistic (x-axis). Here, W explained a 28% of the variation in the absolute value RMSE differences.

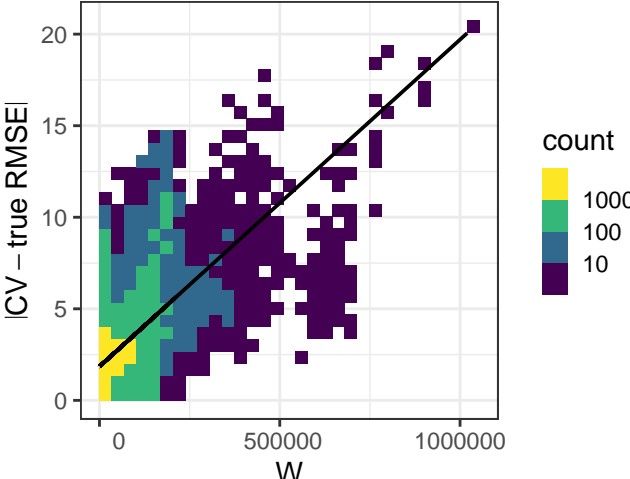

**Figure A4.** The influence of different numbers of k on the difference between the Cross-Validated and true RMSE (upper row), and on the W statistic (lower row). Points indicate median values, while error bars show the first and third quartile. Note that the values of the W statistic were log-scaled.

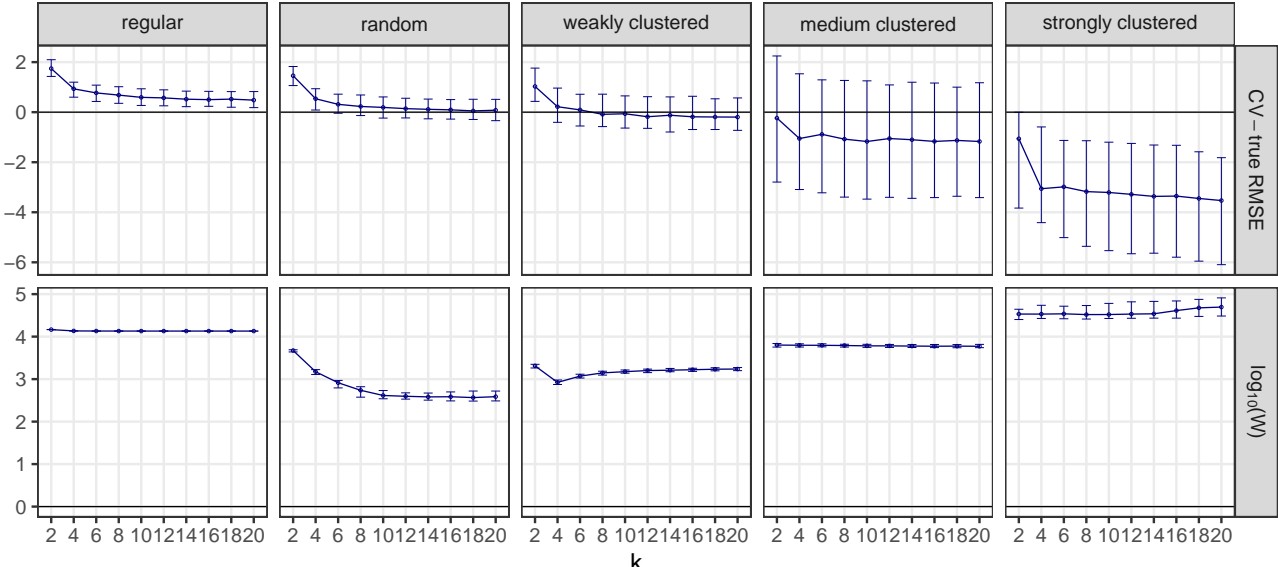

*Author contributions.* All authors conceived the ideas and designed the study. JL and CM developed the algorithm, carried out the analysis, and drafted the manuscript. ML and HM contributed to discussions and drafts. All authors gave final approval for publication.

*Competing interests.* The authors declare no conflict of interest.

*Acknowledgements.* The work was supported by the Federal Ministry of Economic Affairs and Climate action of Germany (project number 335   50EE2009). CM was supported by a PhD fellowship of the *Severo Ochoa Centre of Excellence* programme awarded to ISGlobal.

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
