# Peer review of "kNNDM CV: k-fold Nearest Neighbour Distance Matching Cross-Validation for map accuracy estimation"

_EGUsphere, 2023_

## Author Response (AR1)

Dear Dr Rohitash Chandra,

we would like to thank you for handling the manuscript. Our responses to the reviewer comments and to the community comment are organised in a point-by-point fashion following each comment in italics. The changes made to the manuscript are indicated in blue.

**Reviewer 1**

*The manuscript presents a much-needed methodology for cross-validation of spatial data. In my opinion, the strongest point is the use of the W statistic to identify the best CV split. However, there are a few points which I feel should be addressed in the discussion.*

*The proposed methodology using clustering algorithms seems valid, but how can we know if it provides the best possible result? An algorithm that optimizes the W statistic directly as a function of the CV fold indices would be more desirable, instead of relying on the clustering algorithm´s internal metric as a proxy. As a suggestion for future work, I recommend using a genetic algorithm to assign CV indices to the data points directly.*

We would like to thank professor Gonçalves for his suggestion. Fold assignment via clustering is an intuitive and straightforward approach, however, we agree that using an advanced optimization procedure such as a genetic algorithm might be worth testing. In the revised version of the manuscript we discuss this point as future work, which we will try to take onboard in future versions of the algorithm.

ll. 272-277: Possible future points for investigation regarding kNNDM include a simulation study comparing newly proposed CV-based map accuracy estimation methods (de Bruin et al. 2022) as well as feature-based CV methods (Roberts et al. 2017) in a larger variety of scenarios, also including classification problems, **2) implementing a genetic algorithm that minimizes the W statistic directly as a function of CV folds**, 3) exploring the extension of kNNDM to feature space, and 4) investigating how kNNDM CV can affect feature selection (Meyer et al. 2019), hyperparameter tuning (Schratz et al. 2019), and model applicability (Meyer & Pebesma 2022).

*The W statistic explained 60% of the variability in map accuracy, but would this be consistent across different datasets? At least one more case study would be needed to verify this.*

We agree that a second simulation is helpful to assess if this relationship holds in a different setting. Therefore, we include an additional simulation using the Above-Ground Biomass example described in de Bruin et al. (2022) as supplementary material in the revised version of the manuscript.

**Reviewer 2**

*The study proposes a novel cross-validation method for spatial data that aims to deliver more representative measurements of spatial map accuracy than commonly-used methods. This is a relevant concern for GMD readers with the rise in use of machine learning methods for geoscientific modelling. Issues with model evaluation in the spatial setting have been identified in a number of recent studies. The paper is well-written and contributes a practical solution for a common issue.*

*In my opinion, the most exciting/innovative idea in this work is the concept of defining the evaluation method based on the desired data for which he model is intended to return predictions. This would require researchers to more carefully define the purpose of their models before and during the model*

*creation process, which should be common practice. In reality, this is often not done, or done in a 'standard' way which doesn't accurately reflect the intended use of the model.*

*The method presented in this paper is a very practical solution to this, where the desired target dataset is an input of the evaluation algorithm and therefore researchers are required to clearly consider and define it. I think this is a significant contribution to model development methodology and should be more clearly emphasised in the manuscript. The possibilities, benefits and disadvantages of this concept could also be discussed - for example, when models are used in production, the prediction area is a moving target; would that require continual re-evaluation?*

We thank reviewer 2 for their comments. As discussed in our previous work (Meyer & Pebesma 2022, Milà et al. 2022), we agree with the reviewer on that defining the objective of the prediction is a key step to define an appropriate map accuracy estimation method. In the kNNDM manuscript, we explain this idea in the third paragraph of the introduction and, in the revised version of the manuscript, we also mention it in the concluding paragraph of the discussion.

ll. 284-287: Nonetheless, prediction-oriented CV methods such as NNDM LOO or kNNDM CV that consider the prediction objectives of the model can be used to implement a measure of map accuracy during model development or, in absence of a probability sample, to estimate the map accuracy of a given model.

If the prediction area differs when the model is used for production, the CV estimate might not be a suitable proxy for map accuracy anymore. This would require other testing strategies. Re-evaluation using kNNDM might be one option. We addded a sentence in the discussion section on that issue.

ll. 269-271: Fifthly, the CV error estimate obtained by kNNDM is only reasonable if the prediction area doesn't change when the model is deployed. If the prediction area changes significantly, re-evaluation might be required. Also, when the prediction area is unknown prior to model training, kNNDM cannot be used.

*The paper suggests that kNNDM is, essentially, a computationally-cheaper alternative to the previously-published method by the authors, NNDM LOO. In the article, the only limitation of leave-one-out CV methods described is that of computational time. However, to my knowledge, even if computation is not considered, LOO CV methods may not be the optimum method due to higher variation in the resulting models (due to the bias-variance tradeoff). Could this explain why kNNDM 10-fold seems to perform better in the case of strong clusters (Figure 5)? For me, this would be more convincing than the computation speedup comparison, which is relatively trivial given that LOO CV is the most extreme version of k-fold CV.*

We would like to thank the reviewer for pointing out the bias-variance trade-off in LOO vs. k-fold CV. While in the strong clustered simulations NNDM k-fold CV provides slightly more accurate RMSE (mean (SD) -0.008 (0.043) for kNNDM vs -0.015 (0.038) for NNDM LOO) and MAE estimates (mean (SD) -0.003 (0.031) for kNNDM vs -0.013 (0.026) for NNDM LOO), the difference between the two methods is small and cannot be detected for the $R^2$. Moreover, dispersion estimates for NNDM LOO CV were generally smaller as indicated in the parentheses above. We think that although the two algorithms will tend to provide similar results, differences are expected because the way to match the distributions is different, i.e. a buffering approach is used for the LOO CV while clustering is used for kNNDM. In addition to these, as the reviewer points out, the size of the hold-out data may have an impact on the results as well, with some studies suggesting the aforementioned bias-variance trade-off (Kohavi et al. 1995, Hastie et al. 2009) while others argue

this will actually depend on the modelling algorithm and its stability (Zhang & Yang 2015).

In the revised version of the manuscript, we now discuss how LOO NNDM CV and kNNDM can result in different estimates due to the different methodologies as well as the size of the holdout data in the first paragraph of the discussion.

ll. 225-226: Small differences between NNDM LOO and kNNDM CV can be attributed to the different ways to match the distributions as well as the different hold-out sample size.

*Following on from this, it seems likely that the value of k would impact the results. Use of 10 folds is very common; is there theoretical justification for this? It would be useful to see some comparisons of the results with multiple values of k.*

We chose 10 folds since, as the reviewer points out, it is a frequent choice amongst ML practitioners. However, we agree that $k$ might influence results as it has important implications for Nearest Neighbour Distances (NNDs), e.g. lower numbers of $k$ can better address severe clustering in the data (at the cost of resulting in potentially more biased performance estimates). To address this, in our updated manuscript we have added a new experiment where we investigate the influence of different numbers of $k$ on the performance and quality of the match of kNNDM.

[Figure]

Figure 1: CV error estimates for kNNDM CV with different numbers of $k$ (first three rows). The respective W statistic is shown in the fourth row.

*In Figure 1, it is shown that the W statistic will also be larger if training points are regularly distributed, as well as when clustered. Does this mean that the null hypothesis might be rejected for*

*regularly distributed datapoints? Does this explain why NNDM LOO performed better for regularly distributed data (Figure 5)?*

Regarding regularly-distributed samples in Figure 1, the $W$ statistic between $\hat{G}_{ij}(r)$ and $\hat{G}_j(r)$ for regular samples will indeed be larger since distances between training points will be longer than during prediction (i.e. $\hat{G}_j(r) \leq \hat{G}_{ij}(r)$). This, however, does not impact our results, since the Kolmogorov-Schirnov test we perform as a first step of the algorithm is one-sided and will only perform clustering if the null hypothesis $H_0 : \hat{G}_j(r) \leq \hat{G}_{ij}(r)$ is rejected in favour of the alternative hypothesis $H_1 : \hat{G}_j(r) > \hat{G}_{ij}(r)$, which will not occur for regular samples. As a result, the algorithm will return a random k-fold CV instead.

We think that the worse performance of NNDM k-fold CV compared to NNDM LOO CV for regular samples in Figure 5 is due to the fact that in absence of clustering, NNDM LOO CV generalises to LOO CV while NNDM k-fold CV generalises to random k-fold CV. In a random k-fold CV, neighbouring points can still be in the same fold due to a random chance that will increase with smaller number of folds $k$, thus causing slightly longer NND during CV. In contrast, in a NNDM LOO CV all points except the one being validated will be included in the model leading to the smallest possible W.

*Minor comment: I assume the hyperparameters of the models are not tuned as it is not mentioned, but this could be stated explicitly.*

We confirm we did not tune the model hyperparameters in order to save computation time in the already computationally-demanding simulations, given that our objective was not to optimize the performance of the models. We will add the respective information in the updated version of the manuscript.

ll. 149-150: RF hyperparameters were not tuned and default values were used in all simulations to shorten computation time.

*Finally, I would recommend testing the method on at least one additional dataset, as the results presumably depend on the spatial autocorrelation present in the dataset used.*

We agree with the reviewer that testing the model on a different dataset is needed. We added a second simulation in the revised version of the manuscript where we test the kNNDM method using the Above-Ground Biomass example presented in de Bruin et al. (2022).

**Community comments**

*Just a quick hint that https://doi.org/10.1016/j.jag.2023.103364 was just published - may or may not be relevant for your discussion.*

We thank Dr. Nils Tjaden for providing the reference, which we have included in the revised version of the manuscript.

ll. 263-265: For example, Wang et al. (2023) recently developed a CV method that considers both the geographic and feature space, although it does not consider the prediction domain and predictive conditions of the model.

**References**

de Bruin, S., Brus, D. J., Heuvelink, G. B., van Ebbenhorst Tengbergen, T. & Wadoux, A. M.-C. (2022), 'Dealing with clustered samples for assessing map accuracy by cross-validation', *Ecological Informatics* **69**, 101665.
**URL:** *https://linkinghub.elsevier.com/retrieve/pii/S1574954122001145*

Hastie, T., Tibshirani, S. & Friedman, H. (2009), *The Elements of Statistical Learning. Data Mining, Inference, and Prediction.*, Springer Science & Business Media.

Kohavi, R. et al. (1995), A study of cross-validation and bootstrap for accuracy estimation and model selection, *in* 'Ijcai', Vol. 14, Montreal, Canada, pp. 1137–1145.

Meyer, H. & Pebesma, E. (2022), 'Machine learning-based global maps of ecological variables and the challenge of assessing them', *Nature Communications* **13**(1), 2208.

Meyer, H., Reudenbach, C., Wöllauer, S. & Nauss, T. (2019), 'Importance of spatial predictor variable selection in machine learning applications – moving from data reproduction to spatial prediction', *Ecological Modelling* **411**, 108815.

Milà, C., Mateu, J., Pebesma, E. & Meyer, H. (2022), 'Nearest neighbour distance matching leave-one-out cross-validation for map validation', *Methods in Ecology and Evolution* **13**(6), 1304–1316.
**URL:** *https://besjournals.onlinelibrary.wiley.com/doi/abs/10.1111/2041-210X.13851*

Roberts, D. R., Bahn, V., Ciuti, S., Boyce, M. S., Elith, J., Guillera-Arroita, G., Hauenstein, S., Lahoz-Monfort, J. J., Schröder, B., Thuiller, W., Warton, D. I., Wintle, B. A., Hartig, F. & Dormann, C. F. (2017), 'Cross-validation strategies for data with temporal, spatial, hierarchical, or phylogenetic structure', **40**(8), 913–929.
**URL:** *https://onlinelibrary.wiley.com/doi/abs/10.1111/ecog.02881*

Schratz, P., Muenchow, J., Iturritxa, E., Richter, J. & Brenning, A. (2019), 'Hyperparameter tuning and performance assessment of statistical and machine-learning algorithms using spatial data', *Ecological Modelling* **406**, 109–120.
**URL:** *https://www.sciencedirect.com/science/article/pii/S0304380019302145*

Wang, Y., Khodadadzadeh, M. & Zurita-Milla, R. (2023), 'Spatial+: A new cross-validation method to evaluate geospatial machine learning models', *International Journal of Applied Earth Observation and Geoinformation* **121**, 103364.
**URL:** *https://www.sciencedirect.com/science/article/pii/S1569843223001887*

Zhang, Y. & Yang, Y. (2015), 'Cross-validation for selecting a model selection procedure', *Journal of Econometrics* **187**(1), 95–112.

---

## Author Response (AR2)

Dear Dr Chandra,

thank you very much for your feedback. Our responses to the reviewer comments are organised in a point-by-point fashion following each comment in italics. The changes made to the manuscript are indicated in blue.

**Reviewer 1**

Reviewer 1 did not make any comments.

**Reviewer 2**

*Congratulations on a very interesting paper! The following should be corrected prior to publication (line numbers refer to the track changes version of the paper)*

*Figs 4,5,7, 8 and appendix 2: Change "strongly regular" instead of "strong regular", "weakly regular" instead of "weak regular", "weak clustered" to "weakly clustered", "strong clustered" to "strongly clustered"*

*line 36: "only cover" instead of "only covering"*

*line 43 " this has even been..." instead of "this has been even..."*

*line 56 "reference data randomly distributed within" instead of "randomly distributed reference data within"*

*Line 106 "in the presence" instead of "in presence"*

*line 121 "coordinates of the training points" instead of "training points' coordinates"*

*line 123 "q clusters into k folds" instead of "q into k folds"*

*line 138 " in Euclidean space" instead of " in the Euclidean space"*

*line 139 "as the coefficient " instead of "as coefficient "*

*line 176 either: " a poor estimate" or "poor estimation " instead of "a poor estimation"*

*line 270 "does not" instead of "doesn't"*

*line 286 "in the absence" instead of "in absence"*

We thank professor Mueller for the acknowledgement; we have implemented all suggested edits.

*Finally the figures in appendix 2 need to be improved and titles/ explanations positioned on the same page as the relevant figure*

We have improved the appendix as suggested by the reviewer.

**Reviewer 3**

*This paper is an improvement on the efficiency of the authors' previous work NNDM LOO CV, which is significant from the results (Fig. 8). However, some necessary modifications need to be made before the paper can be accepted. In fact, I don't think there are too many technical problems in this paper, but there is a lot of room for improvement in the writing of the paper. In order to let readers better understand their work and expand the influence of the paper, I suggest that the author consider my opinions.*

Thank you for the acknowledgement and suggestions.

*1) The title of the paper is suggested to be changed to kNNDM CV: k-fold... Or simply remove kNNDM, because k-fold NNDM is more accurate, but this abbreviation is too long.*

We have changed the title to "kNNDM CV: k-fold Nearest Neighbour Distance Matching Cross-Validation for map accuracy estimation" as suggested.

*2) Since the biggest contribution of the paper is to improve the efficiency of NNDM LOO CV method, the abstract needs to explain the specific efficiency improvement shown by the experiment (For example: how many times improved).*

We have added a sentence in the abstract quantifying the efficiency improvement case of 4000 strongly clustered training points (lines 15-18):

We found that kNNDM CV performed similarly to NNDM LOO CV and produced reasonably reliable map accuracy estimates across sampling patterns. However, compared to NNDM LOO CV, kNNDM resulted in significantly reduced computation times. In an experiment using 4,000 strongly clustered training points, kNNDM CV reduced the time spent on fold assignment and model training from 4.8 days to 1.2 minutes.

Furthermore, we have added a sentence in section 3.4 explicitly quantifying the efficiency improvement shown in the experiment (lines 207-209):

For a sample size of 4,000, kNNDM CV reduced the time spent on fold assignment and model training from 3.2 hours to 30 seconds for random samples and from 4.8 days to 1.2 minutes for clustered samples, as compared to NNDM LOO CV.

*3) I noticed that Mila et al., 2022 had been cited 20 times in the paper, which generally only needs to be cited once, not every time it is mentioned.*

We have revised our manuscript and removed instances of the citation that could be omitted. However, there are still multiple citations in the paper that are needed to support certain statements. The total number of references has been reduced to 12.

*4) In Part 2 I noticed that the premise of understanding this paper seems to be reading (Mila et al., 2022), on the basis that I think each paper should be independent. Therefore, I suggest authors supplement the introduction of NNDM LOO method and related concepts in this paper?*

We have reviewed section 2 and added all necessary background information so that reference to Milà et al. (2022) is no longer required to follow the text. Namely, we have:

- Included equations of nearest neighbour distances ECDFs rather than referring to them in the NNDM paper (lines 89, 92 & 98).

- Deleted unnecessary mentions to the original algorithm.

- Modified figure 1 to show an example of k-fold cross-validation rather than a LOO CV.

[Figure]

Figure 1: Top row: prediction points (regular grid) and training points with different spatial distributions (bold), simulated for visualization purposes only. Bottom row: NND ECDF between training and test locations during 10-fold random CV ($\hat{G}_j^*(r, \mathbf{L})$, orange) and NND ECDF between prediction and training locations ($\hat{G}_{ij}(r)$, black) corresponding to each of the sampling distributions in the top row. The shaded grey area corresponds to the W statistic, whose value is printed in the plots.

*5) How does the paper implement k-fold with w statistic? Authors seem to want the reader to understand their thoughts from the algorithm, which is often difficult. I suggest they devote some content to addressing their idea directly. Figure 2 also looks too complicated. The opinions of the given Community comments, I also went to see Wang's paper (https://doi.org/10.1016/j.jag.2023.103364), the paper compared easier to understand, which can be refered.*

We have added a new first paragraph to section 2 which presents, in non-technical language, the objective of the algorithm and the main ideas behind its implementation. With it, we equip the reader with an overall understanding of what we want to achieve before going to the finer and more technical details (lines 77-83):

The objective of kNNDM is to find a k-fold configuration such that the distribution of NND between test and train locations during CV matches as close as possible the distribution of NND between prediction and train locations. In other words, kNNDM aims to create predictive conditions in terms

of geographical distances that resemble those found when using the model to predict a certain area. To do so, we use a clustering approach to create a set of candidate fold configurations with different degree of spatial clustering, of which we choose the one that offers the best match between the two distributions. Before explaining the algorithm in detail, we define the different NND distribution functions used in kNNDM, as well as the statistic used to evaluate the different fold candidates.

Regarding Figure 2, we have carefully considered it and tried to identify elements to simplify according to the reviewer's comment. While we cannot omit any of the included steps as these are the elements of the algorithm, we have reduced the text that was referring to the principal components part. The updated figure looks as follows:

[Figure]

Figure 2: Workflow of the kNNDM algorithm.

In addition to this change, we would like to note that the main part of the algorithm, including the clustering and choosing the best fold configuration by minimizing the Wasserstein statistic, is graphically illustrated in Figure 3. Finally, we refer to Wang's paper in the discussion section (lines 269-271):

For example, Wang et al. (2023) recently developed a CV method that considers both the geographic and feature space, although it does not consider the prediction domain and predictive conditions of the model.

*6) In my opinion, Figure 1 and Figure 3 are also unnecessary. These conclusions can be explained through demonstration or reference. If necessary, they should be included in the experimental part instead of here. The authors should note that since the data is presented in a later section, the reader will have more confusion about how these results were obtained.*

In order to remedy the potential confusion regarding the origin of the data in Figures 1 and 3, we have improved their figure labels to highlight that their use is for illustrative purposes only. The figure captions now read:

Figure 1. Top row: prediction points (regular grid) and training points with different spatial distributions (bold), **simulated for visualization purposes only**. Bottom row: NND ECDF between training locations found during 10-fold random CV ($\hat{G}_j^*(r, \mathbf{L})$, orange) and NND ECDF between prediction and training locations ($\hat{G}_{ij}(r)$, black) corresponding to each of the sampling distributions in the top row. The shaded grey area corresponds to the W statistic, whose value is printed in the plots.

Figure 3. Top row: kNNDM with k=2 (red and blue points) for several number of clusters $q$. Prediction points consist of a regular grid (not shown) spanning the whole polygon. **Points were simulated for visualization purposes only.** Bottom row: NND ECDF between training locations during LOO CV ($\hat{G}_j(r)$, blue), between test and train locations during kNNDM CV ($\hat{G}_j^*(r, \mathbf{L})$, orange), and between prediction and training locations ($\hat{G}_{ij}(r)$, black) corresponding to each CV configuration in the top row.

Regarding moving Figures 1 and 3 to the appendix, we would like the reviewer and editor to consider leaving them in their current placement in section 2. We believe figures 1 and 3 are key visual aides necessary to understand the algorithm being described in that section. Figure 1 illustrates nearest neighbour distance functions as well as the W statistic that have just been described, which may not be clear from the included equations for some readers. Figure 3 complements the text describing the algorithm in that section by showing how the optimal fold configuration is chosen based on the W statistic. We hope that our justification has been convincing enough so as not to alter their placement.

*7) Figure 7. A line chart might be better.*

We have changed Figure 7 and Figure A4 to visualize the data as a line chart, where lines connect median values across different number of folds.

*8) Additional simulation should be placed in the paper, not in a supplement. Figures 1 and 3 can instead be added to the supplement file.*

We now include additional simulations as appendices in the main article. Unlike supplementary material, appendices are included in the main file and thus are accessible to the reader without having to download an additional file. Regarding the suggestion to move Figures 1 and 3 to the supplement, we would like to refer the reviewer to our response to point number *6)*.

[Figure]

Figure 3: CV error estimates for kNNDM CV with different numbers of $k$ (first three rows). The respective W statistic is shown in the fourth row. Points indicate median values, while error bars show the first and third quartile.

**References**

Milà, C., Mateu, J., Pebesma, E. & Meyer, H. (2022), 'Nearest neighbour distance matching leave-one-out cross-validation for map validation', *Methods in Ecology and Evolution* **13**(6), 1304–1316.
**URL:** *https://besjournals.onlinelibrary.wiley.com/doi/abs/10.1111/2041-210X.13851*

Wang, Y., Khodadadzadeh, M. & Zurita-Milla, R. (2023), 'Spatial+: A new cross-validation method to evaluate geospatial machine learning models', *International Journal of Applied Earth Observation and Geoinformation* **121**, 103364.
**URL:** *https://www.sciencedirect.com/science/article/pii/S1569843223001887*